# The influence of visitor-based social contextual information on visitors' museum experience

**Taeha Yi** [1], **Hao-yun Lee** [2], **Joosun Yum** [2], **Ji-Hyun Lee** [2]*

1 Department of Interior Architecture Design, Hanyang University, Seoul, The Republic of Korea, 2 Graduate School of Culture Technology, KAIST, Daejeon, The Republic of Korea

* jihyunlee@kaist.ac.kr

**Data Availability Statement:** All relevant data are within the paper and its Supporting Information files.

**Funding:** This work was supported by a National Research Foundation of Korea (NRF) grant funded

## Abstract

Visitor-centered approaches have been widely discussed in the museum experience research field. One notable approach was suggested by Falk and Dierking, who defined museum visitor experience as having a physical, personal, and social context. Many studies have been conducted based on this approach, yet the interactions between personal and social contexts have not been fully researched. Since previous studies related to these interactions have focused on the face-to-face conversation of visitor groups, attempts to provide the social information contributed by visitors have not progressed. To fill this gap, we examined such interactions in collaboration with the Lee-Ungno Art Museum in South Korea. Specifically, we investigated the influence of individual visitors' social contextual information about their art museum experience. This data, which we call "visitor-based social contextual information" (VSCI), is the social information individuals provide—feedback, reactions, or behavioral data—that can be applied to facilitate interactions in a social context. The study included three stages: In Stage 1, we conducted an online survey for a preliminary investigation of visitors' requirements for VSCI. In Stage 2, we designed a mobile application prototype. Finally, in Stage 3, we used the prototype in an experiment to investigate the influence of VSCI on museum experience based on visitors' behaviors and reactions. Our results indicate that VSCI positively impacts visitors' museum experiences. Using VSCI enables visitors to compare their thoughts with others and gain insights about art appreciation, thus allowing them to experience the exhibition from new perspectives. The results of this novel examination of a VSCI application suggest that it may be used to guide strategies for enhancing the experience of museum visitors.

## Introduction

According to the International Council of Museums (ICOM), an art museum is defined as a space for public education, enjoyment, and the promotion of culture [1]. Museum visitors and their experiences constitute a key component of art museum operations and management; as such, the development of a visitor-centered approach is highly relevant. In applying this approach, attempts have been made to more fully understand the museum experiences of

by the Korea government (MSIT) (No. NRF-2019R1A2C1007042). URL of NRF: https://www.nrf.re.kr/eng/index The funders had no role in study design, data collection and analysis, decision to publish, or preparation of the manuscript.

**Competing interests:** The authors have declared that no competing interests exist.

visitors [2, 3]. Among the various studies, Falk and Dierking [4] proposed a noteworthy approach suggesting that visitors' museum experiences occur amid the interactions of several contexts. Specifically, they defined visitor experience as occurring in the interactions of the physical (i.e., environment, exhibitions, artwork labels, and guide media), personal (i.e., visiting motivation, prior knowledge, and personal interest or choice), and social (i.e., interactions within a group of visitors and conversations with art museum officials or other people) contexts.

Previous research has mainly focused on the interactions among the three contexts mentioned above [5, 6]. However, the interactions between personal and social contexts are not fully researched. Some related studies have focused on communication between accompanying visitors in terms of social contexts [7, 8]. These studies have concentrated on face-to-face conversations between visitors, and there has been insufficient research investigating social interaction from the various visitors to art museums. When forming an opinion in daily life, people are affected by the opinions or reactions of others, which can cause them to change their opinions. Such social information is essential data in almost all product, content, or service areas [9–11]. Through this social process that considers the experiences of ourselves and others together, we can identify the strengths and weaknesses of specific items, further create meaning, and make decisions [12, 13]. However, there have been few attempts to provide supplemental information on visitors' thoughts or reactions to exhibitions in museums and examine the effectiveness of utilizing such information. Regarding supplemental information, the existing research mainly focuses on content written by experts [14, 15]. Thus, to fill the gap in the literature, this study aimed to investigate the influence of the social information contributed by individuals in art museums.

Looking at museums as a service product [16], social information can help to provide a better visitor experience in a social context. To do so, we identified the categories and components of social information via a literature review. Based on the identified social information, we investigated the visitors' needs and opinions to design the museum guidance application and developed an app to deliver the social information. Additionally, we experimented with the influence of visitor-contributed social information using a mobile eye-tracker. In summary, this study attempted to reveal the possibility of enhancing the visitor experience via social information. Further, we proposed a method of providing social information to enrich the visitor experience through museum guidance applications.

## Social information for improving the visitor experience

Researchers have systematically applied advanced digital technology to capture the behavior and reactions of museum visitors to understand their experience [17]. Such studies have aimed to provide researchers or museum experts details about the relationship between visitors' behavior when viewing exhibitions and artworks and their satisfaction [18]. For example, Lanir et al. [19] proposed a system that provides museum experts with visualized visitor behavior information, and Rodriguez-Boerwinkle, Boerwinkle, and Silvia [20] developed the method for art research by tracking visitors in a virtual museum environment. Even though visitor behavior and reaction data have been collected, these collected data have been used as information sources for researchers or museum operators only and have not been provided to museum visitors. Contemporary museums have provided information that meets visitors' desires by applying communicative technology to encourage museum visits [21]. Therefore, we identified the types of data used in prior visitor studies and sought ways to present it as social information that visitors can use to improve their experience in real-time.

Displaying social information is a promising approach for improving certain types of group participation [22], and it has been demonstrated that social information about the involvement of others can be used to encourage user participation and augment their engagement in online communities [23] and physical activities [24]. For example, when added to the context of a player's interactions within an online game, social information can help increase the player's activity [25]. In terms of museums, social information that reveals the presence of others can also be used to enhance visitor participation and broaden visitors' experiences [26].

When studying interactions between personal and social contexts, researchers usually focus on visitor companionship, and social information is mainly based on their communication with accompanying visitors [27]. Although some have tried to use personal information as social information, such as by using SNS in a museum learning context [28] and social tagging systems that gather visitors' thoughts [29], few studies have examined the application of social information data to enhance the visitor experience. To overcome these limitations, we proposed using "visitor-based social contextual information" (VSCI), the social information individuals provide—feedback, reactions, or behavioral data—can be applied to facilitate interactions in a social context. VSCI is a collation of visitors' behaviors and reaction data in museums and provides visitors with information to enhance their museum experience. Therefore, our research questions are as follows:

- RQ 1. What are the components of social information from other visitors in art museums?

- RQ 2. Are visitors' museum experiences enhanced when social information from others is provided to visitors?

## Conceptualization of visitor-based social contextual information

To answer the first research question, we identified the types of data collected from previous visitor studies to determine the information elements constituting the VSCI. We reviewed the previous visitor studies from interdisciplinary research to investigate visitors' behaviors, experiences, reactions, and thoughts systematically. From the literature review, we identified five categories with detailed components. First, visitor evaluation was divided into level of satisfaction with exhibitions and museum objects. Second, visitor behavioral data were collected, such as museum object viewing time, visiting time, and visitor type based on the behavioral characteristics. Third, many studies have paid attention to the visitors' emotions during the art appreciation process. Fourth, some studies focused on museum objects' characteristics to which the visitors reacted sensitively. Lastly, the effect of visitors' comments on their museum experience were analyzed. Subsequently, we established the VSCI with the commonly discussed factors from the literature review, as illustrated in Fig 1.

**Visitor evaluation.** Visitor satisfaction is an important factor that many researchers consider when evaluating museum visit experiences [2, 3, 30]. Han and Hyun [31] demonstrated the importance of measuring museum visitors' satisfaction through empirical studies. In art museum research, the many ways to evaluate aesthetic experiences while viewing artwork are also commonly discussed. Among them, art experience rating scales, such as those measuring visitors' enjoyment, interest, and comprehension, are used frequently [32–34].

**Visitor behavior.** Tracking and timing visitor behavior is the most useful and representative method for understanding how visitors interact with the various elements of an exhibition [35, 36]. Thus, many methods have been used to study visitor behavior, such as capturing visitor location [37, 38] and tracking eye movement [39, 40]. Additionally, valuable measurements are frequently made by tracking and timing visitor behavior in art museums, such as viewing

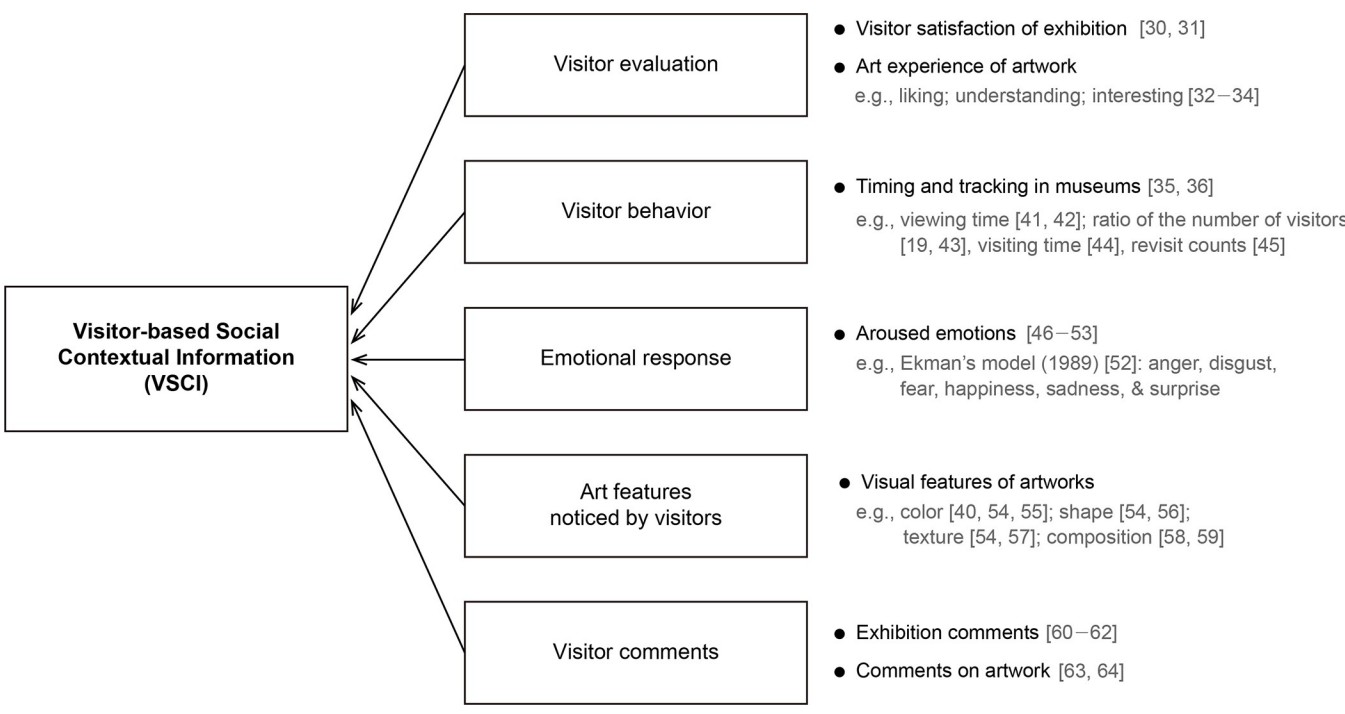

**Fig 1. Conceptual framework of VSCI based on previous visitor studies.**

time [41, 42], the ratio of the number of visitors [19, 43], visiting time [44], and revisit counts [45], for a specific artwork or exhibition.

**Emotional response.**   Emotional response is considered a critical aspect of art experience [46]. Visitors are motivated to find meaningful and personal connections when experiencing emotional reactions to artworks [47]; such connections indicate an enhanced visiting experience. Many researchers have studied the aesthetic emotions of visitors while they appreciate artworks [48]. In these studies, the number of aesthetic emotions noted varies from four [49] to thirty-five items [50]. Additionally, well-known emotional models from Russell [51] or Ekman [52] have often been applied [47, 53].

**Art features noticed by visitors.**   The visual features of artwork are usually discussed in aesthetic experience research. For example, Sartori [54] performed an affective analysis of abstract art based on visual elements (color, shape, and texture), and previous studies have demonstrated that the differences in visitors' art appreciation relate to features such as the color [40, 55], shape [56], and texture [57] of objects. Additionally, Locher [58] revealed that visitors are sensitive to artworks with the right visual composition created by a skilled artist. Similarly, Silvia and Barona [59] demonstrated that visitor preferences are affected by the composition of objects.

**Visitor comments.**   According to Coffee [60], other visitors' comments are important elements that contribute to an individual's museum experience. Comment books can be an essential dialogic activity where social discourse takes shape [61], and they can even stimulate public debate on the visitor experience [62]. Winter [63] also pointed out that although it has been relatively ignored by most museums, the comments of other guests provide visitors with new insights into museum exhibitions. In the field of visitor studies, attempts to analyze visitors' opinions and needs are being made through online visitor reviews [64], which are becoming increasingly important.

## Method

### Study overview

To examine the influence of VSCI on how visitors experience art exhibitions (Fig 2), we defined the elements of VSCI based on commonly discussed aspects of art experience from visitor studies, as follows: visitor evaluation, visitor behavior, emotional responses, art features, and comments. In short, we set the behavior and reaction of museum exhibition visitors as elements of VSCI. Based on those elements of the visitor experience, the research procedure was designed with reference to prototype-based studies that have designed and developed apps or systems by analyzing user requirements [65, 66]. Additionally, an experiment was planned to confirm the differences in visitor experiences depending on whether VSCI was provided or not. Therefore, we designed a research process with three stages: (1) insight research to collect visitors' VSCI needs and opinions; (2) museum application design and prototyping based on the derived insights; and (3) visitor experiments to reveal the impact of VSCI on the visitor experience.

In Stage 1, we used an online survey to investigate visitors' needs and interests to determine the applicable VSCI components. In Stage 2, we developed a prototype of a mobile guidance application to collect and display the VSCI during an individual's visit. Similar mobile guidance apps have become increasingly common for improving the visitor experience in art museum visits [67, 68]; they encourage visitors to stay longer and have a more positive experience than traditional guidance media in cultural exhibitions [69]. Therefore, we conducted visitor observations to collect VSCI on which to base a prototype of a mobile guidance app. In Stage 3, we conducted an experiment with a post-experiment interview to compare the experiences of visitors using the mobile app with VSCI while viewing the exhibition with the visitors' experiences using a mobile app without VSCI. We also tested the differences in visitors' art expertise because previous studies have revealed that visitors' art knowledge and interest influence visitors' behaviors and responses to artworks [70, 71]. To ensure the bioethical integrity of the study, the study design and all the processes of this research were approved by the KAIST Institutional Review Board (NO: KH2019-040).

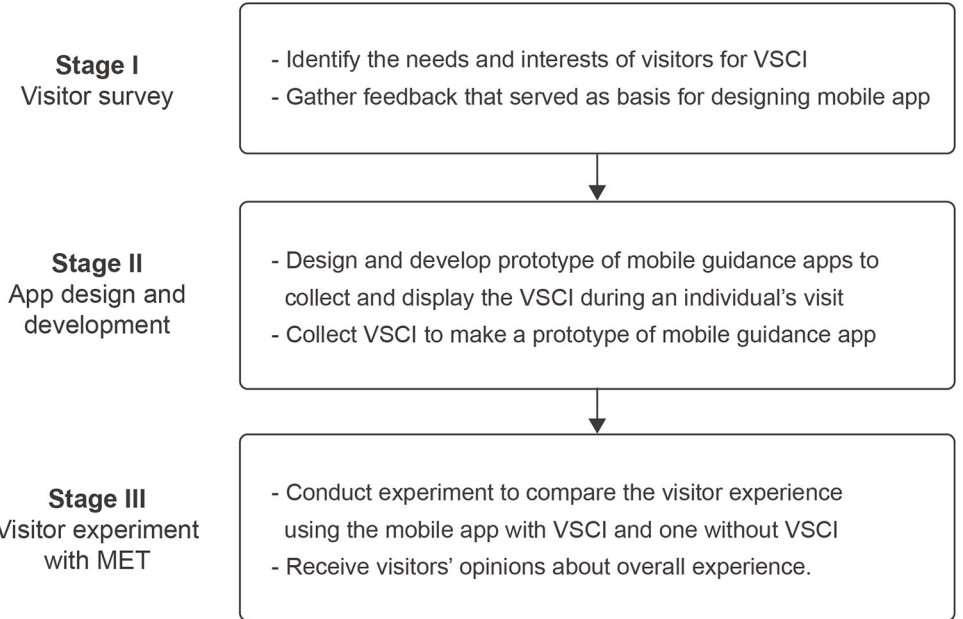

**Fig 2. Overview of research methods.** VSCI = visitor-based social contextual information; MET = mobile eye tracker.

## Stage 1: Online visitor survey to gather feedback for VSCI

We conducted an online survey to understand the interest and needs of visitors for various VSCI factors; the survey content was compiled based on previous visitor studies. An online survey was chosen to obtain opinions from the various visitors who had experience visiting art museums and using mobile guidance services. The survey responses were collected via Google Forms from May 24 to June 8, 2021. We applied the survey data to calculate the importance ranking according to visitors' "degree of curiosity" about VSCI factors. Additionally, we conducted k-means clustering to investigate grouping characteristics according to the ratings for the VSCI elements. The survey results were then used to design a VSCI mobile application and recruit participants for an experiment.

**Participants.** A total of 71 participants (female: 42, male: 29) took part in the survey, and the average age was 31.02 (SD = 5.18) years. Those who participated in this survey received a USD 2 reward. The participants included university students, office workers, and museum workers recruited from multiple online communities in South Korea (e.g., ARA and Hongik-in). Among the participants, 30 were art majors, and 41 were non-arts majors.

**Materials.** The survey we used, a copy of which is provided with additional notes in S1 Appendix, consisted of four sections. The first section included questions for collecting participants' demographic information and asked participants if they were willing to participate in future visitor experiments. The second section gathered information regarding what the participants were curious about when viewing artworks and exhibitions. This section included a total of 17 information elements, for which the degree of participant interest was measured using a 7-point Likert scale (1: very negative to 7: very positive). They were also asked to write down the reason for giving the score and any additional elements about which they were curious. The third and fourth sections related to the participants' art interest [72] and art knowledge (similar to Belke, Leder, and Augustin [73]); this information was intended to allow us to determine if the distinction between art majors and non-majors makes a difference in their levels of interest or knowledge.

## Stage 2: Design and development of mobile guidance application prototype

We considered the importance level of each VSCI element based on the survey outcome and the overall visit process to design the interaction flow, as illustrated in Fig 3. The importance ranking of VSCI elements from the survey helped clarify the information hierarchy used in the mobile applications interface design. According to Djamasbi and Hall-Phillips [74], an interface with a clear hierarchy provides a better experience than one that has no difference in the relative importance of page elements.

In addition to the importance ranking, it is necessary to define some of the contents of the VSCI before designing the application, including emotional response and art features. Since cognitive load is an important factor when designing interfaces [75], and working memory is limited by the number of items [76], we minimized the number of items on the interface to reduce the cognitive load. Specifically, in our prototype, we used Ekman's model [52] with six simple emotions to indicate the visitor's emotional response to an artwork. The six basic cross-cultural emotional responses included anger, disgust, fear, happiness, sadness, and surprise. An additional item, "indifference," was added to represent neutral responses [47]. Moreover, based on the abovementioned art features from previous research, four components were set as options for the visitor to select the visual elements of the artwork they saw: color, composition, shape of the objects, and texture.

Furthermore, we used the average behavioral data to recognize the visitor types derived by comparison with others. This comparison relies on the ability to track the behavior of visitors

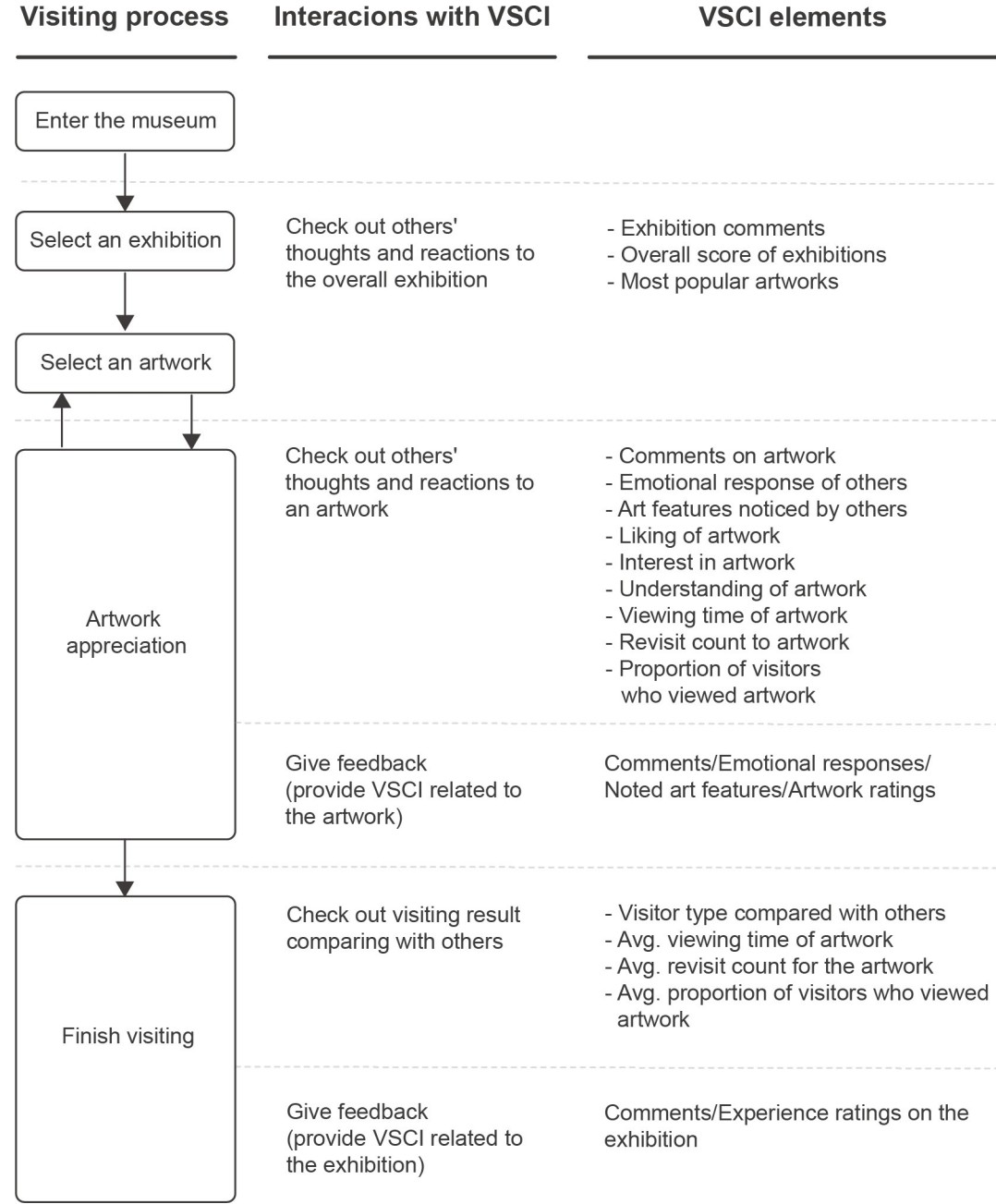

**Fig 3. The process of mobile guidance application with visitor-based social contextual information (VSCI).**

in real-time. Thus, we determined our prototype would include a built-in function that records and calculates the app log data from visitors. For example, the visiting time was measured from the log-in time to the log-out time, and the viewing time was calculated between the entering and exiting time points to and from the specific artwork pages. Additionally, the "revisit count" for each artwork was calculated using the number of times each visitor approached and temporarily remained in front of the artwork. Next, after identifying these factors as four levels based on quantiles, the sum of the levels for each visitor was calculated. Finally, the application was designed to capture the visitor type with three equally sized groups,

namely, diligent (high-rank), selective (middle-rank), and busy visitor (low-rank), referring to Sparacino [77]. Additionally, we set up "most popular artworks" in the app to display all of the works whose scores were within the top 30% for visitors' "likes." The remaining artworks were considered less popular.

To develop a prototype mobile application, we selected artist Ung-no Lee, a master of South Korean contemporary art, who creates modern abstract paintings using traditional Oriental art ink. Since visitors tend to prefer works of famous artists and more frequently judge them as beautiful [33], we planned an exhibition consisting of works of only one artist to eliminate the effect of the artists' fame. In collaboration with the Lee-Ungno Art Museum, we selected five series representing all his artworks, with four artworks in each series. In this collaboration, we designed a simple exhibition called "The Art World of Lee Ung-no" that introduced pieces representative of the artist's work. We developed the mobile application using the Python (v.3.9) Flask web server platform (v.2.0.1), HTML, and JQuery.

## VSCI preparation for the prototype

To make the prototype for the experiment, we needed to prepare real VSCI beforehand. There-fore, we conducted visitor observations to collect data with participants recruited from those who participated in our survey (randomly selected, n = 10). The observation simulated the experiment's main process and environment to collect the participants' feedback, reactions, and behavioral data as VSCI. The participants were asked to provide VSCI for every artwork and exhibition (Table 1). However, VSCI from other visitors was not displayed to the individ-ual visitor participants during the process.

To collect visitor behavioral data, we measured eye-tracking data through Pupil Capture v.3.4 software. We set each surface (area of interest [AOI]) in advance by attaching markers on four sides of the displayed artworks. The viewing time (VT) for each AOI was measured by cal-culating the timestamps of all events where the gaze point of the participant entered (enter time) and exited (exit time) the AOI. The visiting time was measured as the sum of overall viewing time (VT_O) by adding the VT for a set of AOIs:

$$VT = \sum_{i=1}^{n}(Exit\ time_i - End\ time_i)\ in\ AOI \tag{1}$$

$$VT\_O = \sum_{i=1}^{n}(Exit\ time_i - End\ time_i)\ in\ set\ of\ AOI \tag{2}$$

**Table 1. VSCI elements to which participants responded in the observation process.**

| Case | Question |
| --- | --- |
| After viewing an artwork | 1. Leave your comments about this artwork. |
| | 2. Which emotions did you feel were aroused by this artwork? |
| | (1) Sadness; (2) Happiness; (3) Surprise; (4) Fear; (5) Anger; (6) Disgust; (7) Indifference |
| | 3. What did you pay attention to on this artwork? |
| | (1) Color; (2) Form/Shape; (3) Composition; (4) Texture |
| | 4. Please rate this artwork. (from 1 star: *negative* to 5 stars: *positive*) |
| | (1) Interest; (2) Liking; (3) Understanding |
| After viewing the exhibition | 5. Please leave your comments about this exhibition. |
| | 6. Please give an overall score for your opinion about this exhibition. (1 star: *negative* to 5 stars: *positive*) |

VSCI represents "visitor-based social contextual information".

The revisit count for the artwork (RC_A) refers to the number of times a visitor returned (at least once) to see a specific artwork. We also counted the number (CN_A) of cases where the participant's gaze moved to another work after viewing a specific work. When we collect the CN_A for each artwork and exclude the first viewing situation, we can get the revisit count as follows:

$$RC\_A = \sum_{i=1}^{n}(CN\_A_i - 1) \ in \ AOI \tag{3}$$

In addition, when we divide the total number of visitors by the number of visitors who have the value of CN_A, the proportion of visitors who viewed a specific artwork (PV_A) can be measured, as below:

$$PV\_A = \frac{Number \ of \ visitors \ with \ CN\_A}{Total \ number \ of \ visitors} \tag{4}$$

## Stage 3: Visitor experiment

We designed an experiment to determine the influence of providing VSCI to visitors by examining visitors' behavior and reactions. For tracking and timing visitor behavior, indoor positioning technologies can be used; some examples are beacons [78], mobile sensors [79], and cameras [80]. However, these methods make it difficult to accurately determine whether the visitor is viewing the artwork or the mobile phone. In other words, it is difficult to distinguish between and measure the amount of time spent looking at the mobile app while in front of each exhibit. However, in this study, it was necessary to distinguish accurately when a visitor is viewing an artwork or the mobile app; therefore, we used a mobile eye-tracker (Pupil Labs, Pupil Core 120 Hz binocular) to collect the behavioral data of visitors. Mobile eye-tracking (MET) technology is becoming a widely used tool for understanding visual processing and visitor behavior [81]. It also provides rich data through a scene camera that allows users to obtain information about the environment [82].

**Participants.** Forty participants (mean age = 27.29; SD = 5.02) were recruited for the experiment and received a USD 10 reward. All participants had normal or corrected-to-normal vision and did not wear glasses for mobile eye-tracking. We contacted those who agreed to participate in this experiment via the online survey. The participants were recruited based on their gender (male:female = 19:21), major in art or not (art major:non-art major = 16:24), and clusters derived from the online survey (Cluster 1:Cluster 2 = 20:20).

**Materials.** The two types of "Art World of Lee Ung-no" exhibitions were displayed in a laboratory with three white walls, decorated to resemble the environment of a real museum. Some studies have found differences in art appreciation depending on the size and location of the artworks [83, 84]; therefore, all artworks displayed were similar in terms of size (fitted to A3 size), spacing between artworks, and installation height on the wall. We also used high-quality images of artwork provided by the Lee-Ungno Art Museum, given that existing research indicates no significant difference in the appreciation of original artworks [85]. Simple descriptions of each artwork were similarly designed (i.e., word count: $M$ = 24.85, $SD$ = 1.45) and content (series explanation and image interpretation).

Two curators whose majors were in art planning were consulted in preparing the descriptions. To determine the impact of VSCI in a visitor experiment, we divided the art series equally into two exhibitions so that a visitor could appreciate the two exhibitions (E1 and E2) comprising different artworks. Next, we prepared apps for the four conditions by developing apps with a 2 (with or without VSCI) × 2 (E1 or E2) design. Finally, to minimize the differences attributable to the mobile environment (e.g., screen size and speed performance), all visitors were asked to use the same mobile phone (Samsung Galaxy Note 9).

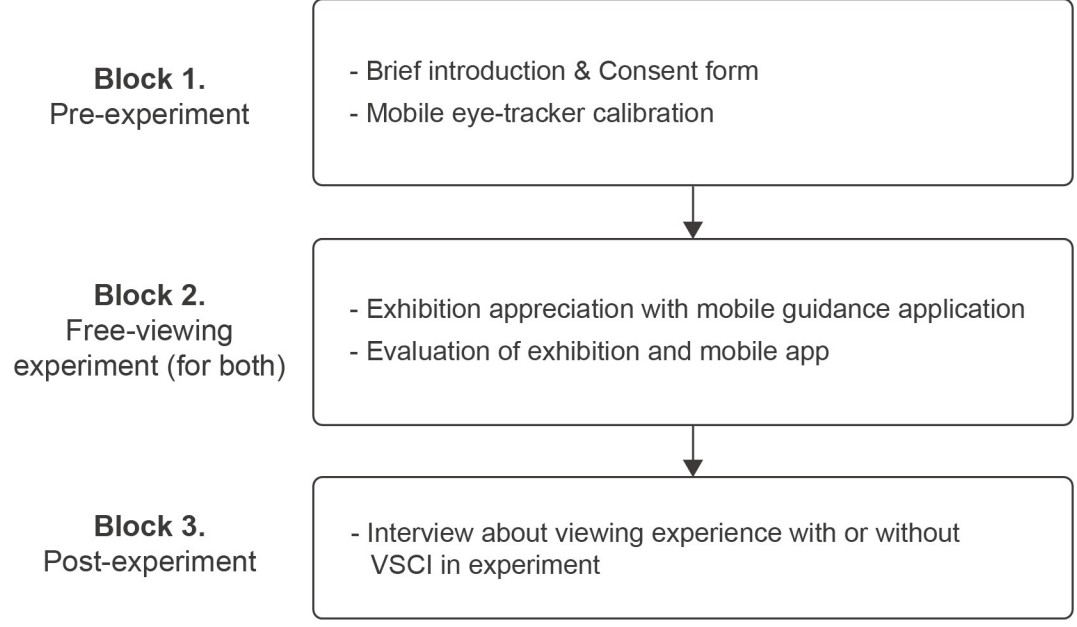

**Block 1.**
Pre-experiment
- Brief introduction & Consent form
- Mobile eye-tracker calibration

**Block 2.**
Free-viewing experiment (for both)
- Exhibition appreciation with mobile guidance application
- Evaluation of exhibition and mobile app

**Block 3.**
Post-experiment
- Interview about viewing experience with or without VSCI in experiment

**Fig 4. Experimental process for the three blocks.** VSCI represents "visitor-based social contextual information".

**Procedure.** This experiment was conducted from August 6 to August 21, 2021, as three blocks (Fig 4) and took about 90 mins, on average. In the first block (pre-experiment), the experiment was introduced to the participants, and their written consent forms were collected. Next, the participants were asked to wear the MET and look at the nine marker points for calibration. Finally, to check the calibration accuracy, the experimenter drew a large circle with their index finger in front of the participant and confirmed whether the participant's gaze followed the position of the fingertip correctly.

In the second block, the participants freely viewed the exhibition under two conditions comprising different artworks (E1 or E2) and mobile apps (with or without VSCI) to eliminate the effect of viewing the same materials. For example, if P01 viewed the first exhibition using a mobile app with VSCI in the first session, they viewed the second exhibition using the mobile app without VSCI. All participants were randomly assigned to a sequence of viewing cases, and 20 cases were performed for each condition. After each viewing case was finished, the participants were asked to evaluate their satisfaction with the current exhibition and the mobile app.

In the third block, after finishing the free-viewing block, visitors were asked to move to an interview space. In the interview, we asked questions about the participant's overall experience of viewing the exhibitions and of the mobile app (e.g., "Was there any difference in the exhibition experience between the two mobile guidance applications?" "Which of the two mobile applications do you prefer? What is the reason?"). Using a 5-point Likert scale, we also collected the preference level ("How much do you prefer this information element?") and the participants' opinions ("Why do you prefer (or not) this information element?") for each component of the VSCI.

**Data analysis.** This study analyzed visitor behavior data and satisfaction with exhibitions and mobile apps to determine whether there is a difference in their exhibition experience depending on the VSCI. To analyze the visitor behavior data, we collected the viewing time of AOI in the same ways as the VSCI preparation phase. In the experimental environment, to

measure the viewing time of the artworks and the mobile app, the mobile phone was also set as AOI by attaching markers, and the viewing time of the artwork (VT-A) and the mobile app (VT-M) was measured separately. The visiting time was calculated by adding both the time to view the artworks and the apps (VT-O = total of VT-A + total of VT-M).

Next, through the interview, we measured exhibition satisfaction using three items from Han and Hyun [31] and the mobile app satisfaction using four items from Song et al. [86] on a 5-point Likert scale (see Table A in S2 Appendix). Lastly, we conducted multiple analyses of variances (ANOVAs) on the difference between the participants' behavioral data and satisfaction according to the experimental conditions. For all measurements and statistical analyses for this study, we used Python version 3.9 and the python libraries "statsmodels (v.0.13.0)," "researchpy (v.0.3.2)," and "scipy (v.1.7.1)".

## Results

### Online survey

In the survey, we asked about participants' needs for information elements when viewing artworks and viewing an exhibition (Table 2). We found that the survey participants were interested in the comments on artwork and emotions of others, as well as an artwork's notable features among the elements of VSCI. They were also interested in knowing which artworks were most popular, the comments related to visitors' impressions of the exhibition, and their visitor type compared with others.

**Differences by art expertise.** In sections three and four of the survey, we checked whether there was a difference between art majors and non-majors with regard to art knowledge (maximum 100 scores) and interest in art (maximum 77 scores). Regarding art knowledge, the average score for the art major group was 59.33 ($SD$ = 18.13), and that for the non-art major group was 26.71 ($SD$ = 8.26). Regarding the degree of interest in art, the average score for the art major group was 52.17 ($SD$ = 14.46), and that for the non-art major group was 37.37 ($SD$ = 13.48). There were statistically significant differences between the two groups in both

**Table 2. Results of the ratings of participants on the elements of visitor-based social contextual information.**

|  | Factor | Mean (SD) |
|---|---|---|
| Viewing Artwork | Comments on artwork | 5.25 (1.54) |
|  | Emotional response of others | 5.00 (1.86) |
|  | Artwork features from others | 4.97 (1.72) |
|  | Interest in artwork | 4.54 (1.90) |
|  | Understanding of artwork | 4.49 (1.89) |
|  | Like the artwork | 4.35 (1.97) |
|  | Viewing time of artwork | 3.90 (1.81) |
|  | Proportion of visitors who viewed artwork | 3.62 (1.86) |
|  | Revisit count for artwork | 3.44 (1.69) |
| Viewing exhibition | Most popular artworks | 5.38 (1.77) |
|  | Exhibition comments | 4.97 (1.80) |
|  | Visitor type compared with others | 4.80 (1.98) |
|  | Exhibition satisfaction | 4.66 (1.77) |
|  | Less popular artworks | 3.83 (2.08) |
|  | Average viewing time of artworks | 3.77 (1.80) |
|  | Average proportion of visitors who viewed artworks | 3.65 (1.90) |
|  | Average revisit counts for artworks | 3.42 (1.70) |

sections (art knowledge: $t(37.85) = 9.182$, $p<0.001$; art interest: $t(60.01) = 4.385$, $p<0.001$). In other words, there was a clear difference in art expertise between art and non-art majors. Next, we looked at the differences between the two groups for all VSI elements and found that the groups did not differ significantly in all elements (all $p>0.05$). These results indicate no significant difference in the needs for VSCI elements based on art expertise.

**Visitor grouping through K-Means clustering.** After confirming that there were no differences in the VSCI needs based on art expertise, we examined whether differences can be found among the groups when using the k-means clustering technique. The results from clustering ($k = 2$) demonstrated differences in all the VSCI elements between the two clusters (Fig 5). The first cluster demonstrated a positive trend of over 4 points (4 points is *neutral* on the 7-point Likert scale) and high average scores in all VSCI elements compared with the second cluster. For the second cluster, only one item ("comments about an artwork") was over 4 points, while the others had a negative trend of fewer than 4 points. Next, an ANOVA was performed to check the differences between the two clusters for all VSCI elements (all $p$s$<0.001$). No differences were found in gender, age, and art expertise between the two clusters.

Considering the results, we examined the participants' answers regarding their interest in the VSCI information. For the first cluster (n = 45), the main reason participants found certain elements interesting is captured in P13's response: "I enjoy comparing other people's ideas to mine." This opinion was shared by 66.7% of the participants. Meanwhile, for the second-largest cluster (n = 26), was the group identifying the reason captured in P03's response: "I do not feel the need to compare my feelings and others because my personal feelings and others' feelings are a unique experience" was reflected in the responses of 69.2% of the participants. These results might be interpreted as a difference in personal interest in the information of others. In conclusion, we did not find any differences in information needs based on art expertise. However, a clear difference was found between those who were curious about others' reactions and those who were not.

## Prototype of a mobile guidance application

Based on the survey outcome, we displayed each VSCI element considering the importance level of the elements. Additionally, we completed the prototype by inputting the VSCI collected through the data preparation phase into the mobile app, as illustrated in Fig 6. The developed mobile guide application with VSCI comprised: (1) a simple description of the ongoing exhibition with visitor satisfaction and comments; (2) indicators of popular artworks such as star markings based on the "liking" scores of visitors; (3) visitor comments on an artwork; (4) reaction tabs about the artwork, which contain the emotion, art features, ratings, and behavioral data from visitors; and (5) visitor type through a comparison with the visitors' behavioral data. The app without VSCI comprised exhibition information, locations of exhibited works, and descriptions of artworks, similar to the existing guide app.

## Visitor experiment

First, we confirmed whether there were statistical differences based on the demographic information and experimental environments. There were no significant differences by gender (male or female), major (art or non-art majors), experimental environments (first exhibition [E1] or second exhibition [E2]), and exhibition appreciation sequence (first viewing or second viewing) in terms of exhibition satisfaction, app satisfaction, and visiting time. Next, we analyzed the differences in the behavior and satisfaction of visitors according to the experimental conditions (Table 3).

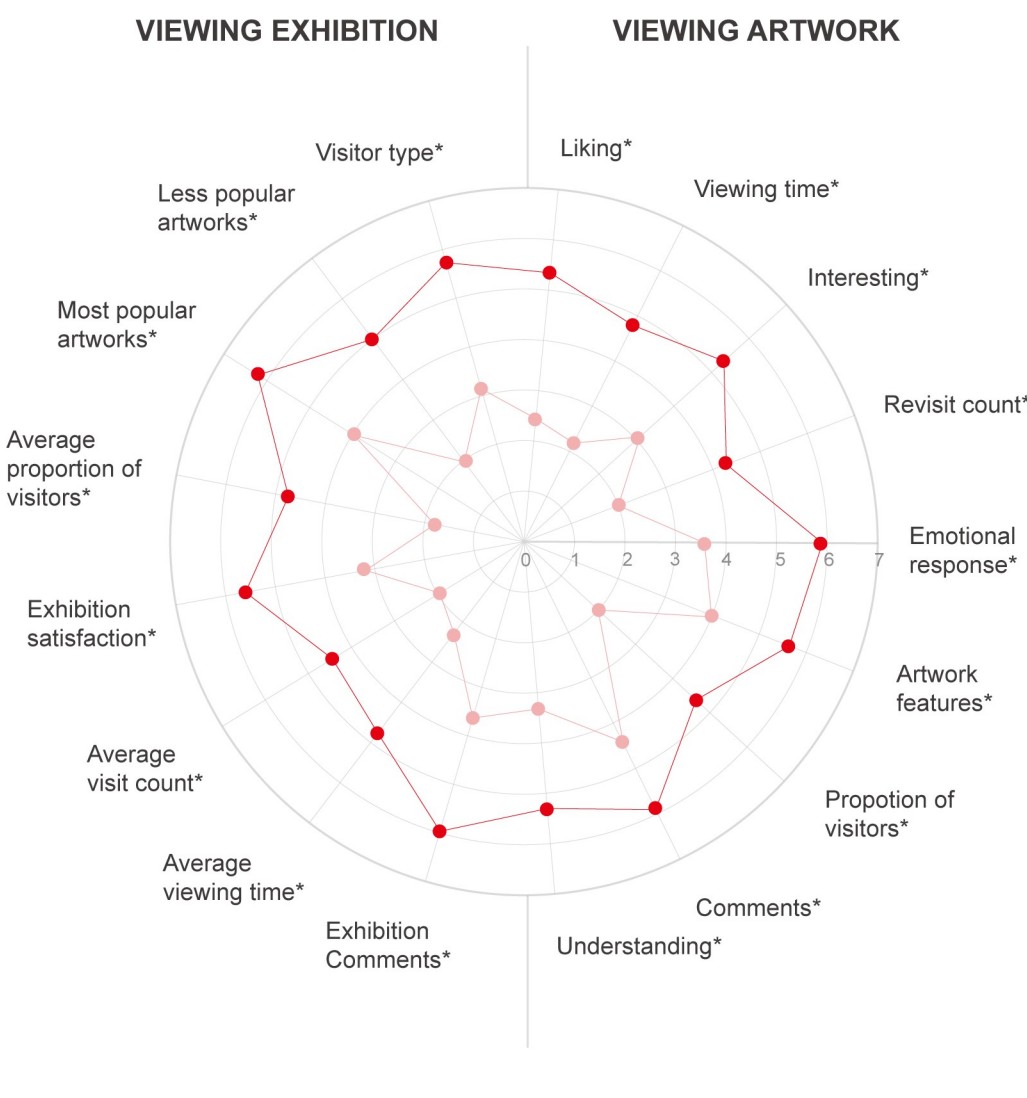

Fig 5. Results of k-means clustering through an online survey analysis.

To explore the effect of VSI, we conducted separate ANOVAs with post hoc Tukey honestly significant difference (HSD) corrections for each factor (Fig 7). Visiting time differed significantly between conditions ($F(3,76) = 6.40$; $p<0.001$). Compared with viewing exhibitions without VSCI, participants viewing exhibitions with VSCI spent more time visiting the exhibition ($Mean_{diff}(cond1, cond2) = 13.98$ min; $Mean_{diff}(cond3, cond4) = 13.56$ min.) There was also a statistically significant difference in viewing time for the mobile app ($F(3,76) = 5.32$; $p = 0.002$). However, in the case of viewing time for artwork, there was no significant difference ($F(3,76) = 2.35$; $p = 0.08$), and no differences among conditions were detected using a post hoc Tukey test (Fig 7B). This implies that if VSCI is given, viewing time for mobile apps could increase, but not the time of viewing artworks. This result is similar to Temme [87], who indicated that although audiences want more information related to artwork, providing more information does not increase their viewing time. In addition, Temme [87] demonstrated that as the amount of information increases, the enjoyment of artworks decreases, and there is no

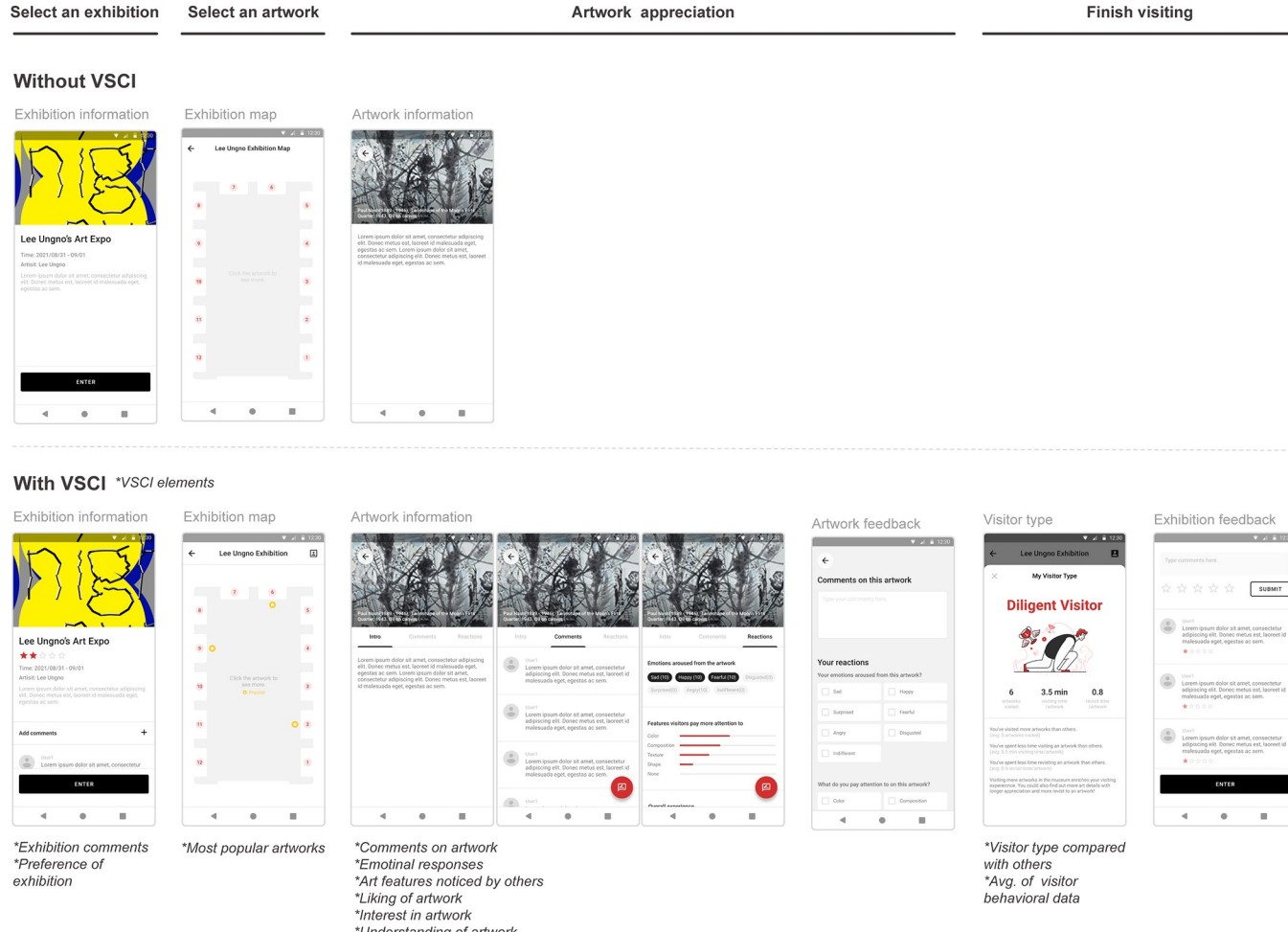

**Fig 6. Prototype of mobile guidance applications (two types of apps: With VSCI and without VSCI).** VSCI represents "visitor-based social contextual information".

**Table 3. Mean (and standard deviations) in exhibition appreciation divided by conditions.**

| | Factor | Condition 1: E1 with VSCI (n = 20) | Condition 2: E1 without VSCI (n = 20) | Condition 3: E2 with VSCI (n = 20) | Condition 4: E2 without VSCI (n = 20) |
|---|---|---|---|---|---|
| Visitor behavior | Visiting time (min.) | 38.92 (18.37) | 24.94 (13.89) | 37.54 (16.94) | 23.98 (10.04) |
| | Viewing time for artwork (min.) | 24.07 (11.90) | 17.73 (12.12) | 22.42 (14.71) | 15.40 (6.95) |
| | Viewing time for mobile app (min.) | 14.85 (11.75) | 7.21 (5.05) | 15.12 (7.33) | 8.58 (6.32) |
| Visitor evaluation | Exhibition satisfaction (5-Likert) | 4.03 (0.53) | 3.08 (0.56) | 4.08 (0.39) | 3.28 (0.73) |
| | Mobile app satisfaction (5-Likert) | 3.76 (0.52) | 2.80 (0.93) | 3.90 (0.48) | 2.86 (0.95) |

E1 = First exhibition, E2 = Second exhibition, and VSCI = visitor-based social contextual information.

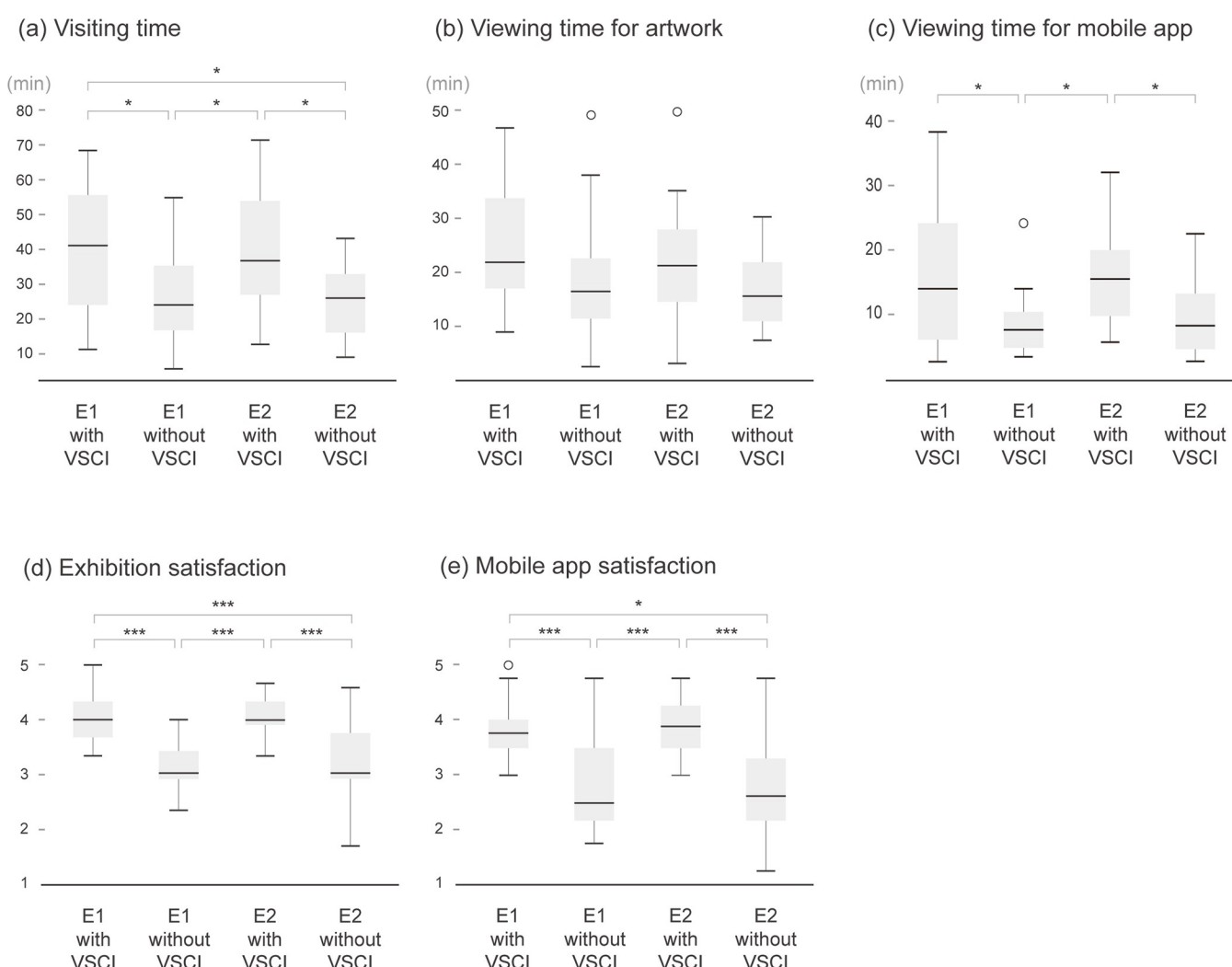

**Fig 7.** Box-and-whisker plots of the four conditions for visitor behavior (A to C) and satisfaction (D, E). The results of multiple comparison corrections by Tukey HSD between groups are presented (* denotes $p<0.05$; **, $p<0.01$, and ***, $p<0.001$). E1 = first exhibition, E2 = second exhibition, and VSCI = visitor-based social contextual information.

difference in interest in art labels. On the contrary, when VSCI was presented, both the satisfaction of the exhibition and mobile app increased. Moreover, the increase in visiting time when VSCI was provided suggests the possibility of offering visitors an opportunity to enjoy an exhibition for a longer time without shortening the time of viewing artworks.

Visitor satisfaction was high when VSCI was provided according to both indicators (exhibition satisfaction: $Mean_{diff}(cond1, cond2) = 0.95$, $Mean_{diff}(cond3, cond4) = 0.8$; mobile app satisfaction: $Mean_{diff}(cond1, cond2) = 0.96$, $Mean_{diff}(cond3, cond4) = 1.04$), and the difference was significant (exhibition satisfaction: $F(3, 76) = 16.47$, $p<0.001$; mobile app satisfaction: $F(3, 76) = 11.88$, $p<0.001$). The result of the post hoc multiple comparisons demonstrated obvious differences between four conditions (Fig 7D, 7E). In other words, the provision of VSCI to visitors appeared to have a positive effect on exhibition appreciation. Moreover, in the post-experiment interview, most of the participants (87.5%) answered that they preferred the application with VSCI. They mentioned that the additional information provided by the app increased their interest and understanding when viewing the artworks and exhibitions. Only one visitor preferred the app without VSCI because they did not want their personal viewing

to be affected by others' reactions (P3, "I don't like being influenced by other people's opinions"). The rest of the visitors (10.0%) mentioned that the preferred app differs depending on the situation: (1) whether they have ample time (e.g., P37, "I think I will use the app with VSCI if I have enough time to enjoy the exhibition") and (2) whether visiting with a companion (e.g., P26, "If I have a companion, I will probably use the app without others' information").

Additionally, a Pearson correlation coefficient was computed to assess the linear relationship between exhibition satisfaction and mobile app satisfaction and demonstrated a high positive correlation between the two variables ($r = 0.71$, $p < 0.001$). This result implies that visitors' satisfaction with the mobile app affects their exhibition experience. In summary, the results illustrate that in most cases, providing mobile guidance applications with VSCI has a positive effect on the exhibition experience of visitors.

**Evaluation of VSCI elements.** The visitor evaluation of the detailed VSCI elements was analyzed, as illustrated in Fig 8. Overall, "Visitor type ($M = 4.25$; $SD = 0.81$)" and "Comment on artwork ($M = 4.25$; $SD = 0.87$)" were evaluated as the most positive factors, with few responses in the negative region (1 to 2 points). By contrast, "Viewing time of artwork ($M = 3.03$; $SD = 1.27$)," "Emotion of artwork ($M = 3.0$; $SD = 1.1$)," and "Proportion of visitors to artwork ($M = 2.98$; $SD = 1.25$)" were rated relatively negatively, as the rating distribution appeared evenly.

During the post-experiment interview session, we collected the opinions of the participants on each VSCI element. We analyzed the interviews to identify the key features, among which the comparison with others' responses was most evident in "Visitor type" and "Comments on artwork." Regarding "Visitor type," the participants enjoyed knowing their type and visiting behavioral characteristics based on a comparison with others (e.g., P12, "I think it is fun to show my type like MBTI. I think the comparison information with other people helped me better understand myself"). Regarding the "comments on artworks," most participants thought that it was good to be able to compare their own thoughts with the opinions of other people. In the process of comparison, they found pleasure in reading the comments that expressed thoughts they shared (e.g., P27, "It was nice to see what people think and see if they have the same thoughts as me"). They also gained a new perspective on an artwork from the comments of others (e.g., P36, "I was looking at an artwork and thought it was not good because it was too dark. However, after reading the reviews saying that it was like a star in the dark night sky, the artwork looked new"). Through social comparisons, negative reactions to information that were inconsistent with their thoughts arose (e.g., P18, "There were a lot of different opinions from mine, so it didn't really touch me"). Specifically, regarding the "Most popular artworks," negative opinions were expressed, and the participants were uncomfortable knowing that the artworks they liked and the displayed artworks were different.

Next, the participants mentioned that the VSCI helped them understand the exhibition or artworks and determine their viewing priorities regarding exhibition content. Regarding the "Most popular artworks," there were positive opinions, such as that VSCI gave some clues as to which artworks to look at in detail and that they can appreciate thinking about what makes a specific artwork popular. Moreover, some participants said that numerical information, such as behavioral data, was helpful for judging the artworks objectively (e.g., P11, "I think numerical information is useful when deciding which artworks to see in detail"). This relates to Screven's [88] results, who reported that it is necessary to provide information to address the problem of visitors struggling to determine why the artworks exhibited in art museums are important to them. In other words, it implies that VSCI, which represents the others' art experience, can play a role in improving the understanding of artworks.

Lastly, for all VSCI elements, most of the differences were in the gap between those who wanted to be provided with the information element and those who did not. As various VSCI

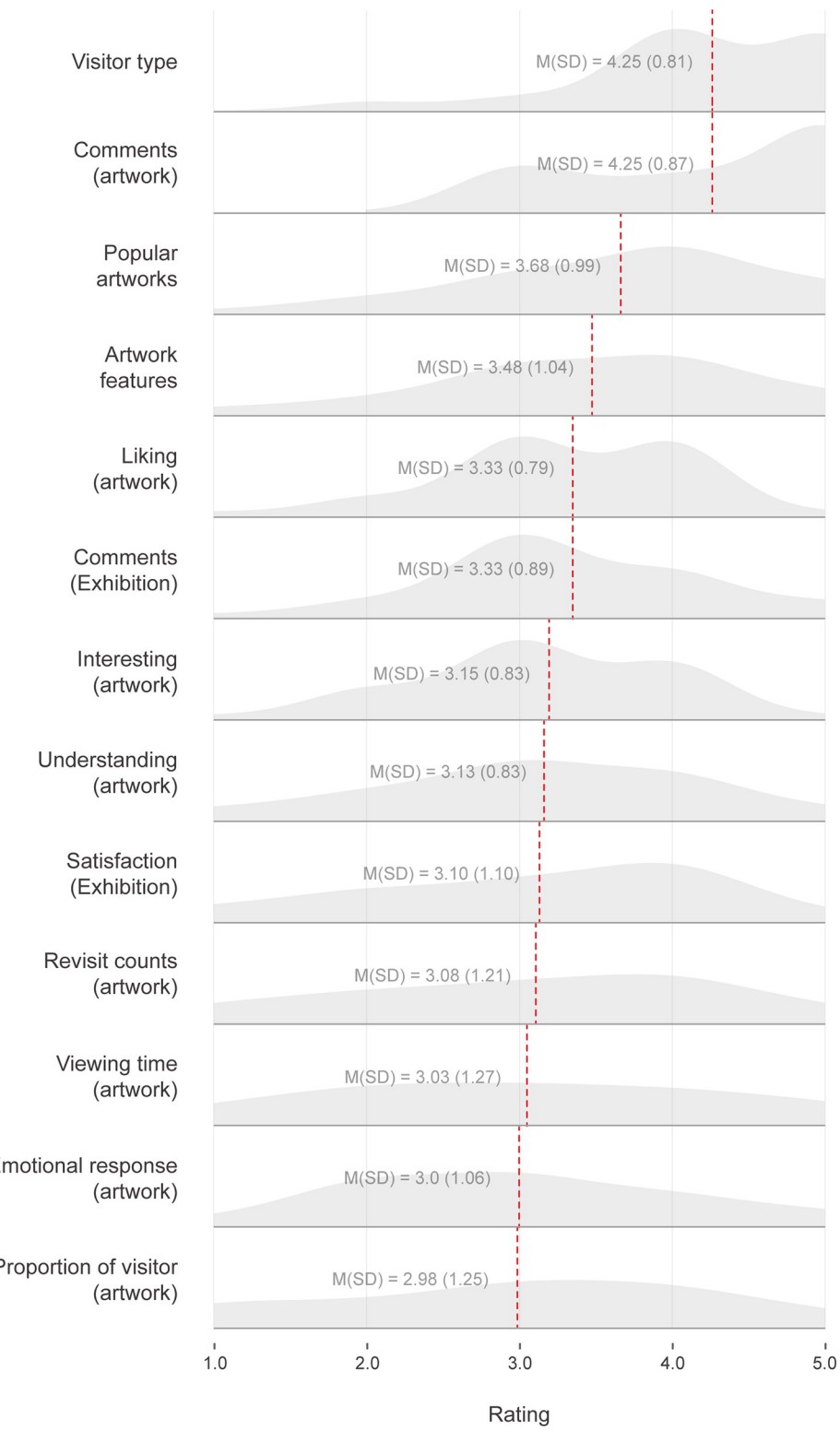

**Fig 8. Ridge plot of visitor ratings for VSCI elements (drawn in the order of average scores [highest to lowest], depicted by the red lines).** VSCI represents "visitor-based social contextual information".

elements were provided in this study, the visitors' needs were also widely revealed. Therefore, to reflect the various needs of these visitors for VSCI, it is possible to apply a customization option that allows individual visitors to manage information elements. For example, in the case of a visitor who does not want their personal viewing to be disturbed by the VSCI (e.g., P19, "Information about how other people rated the exhibit was meaningless and disturbing"), it is possible to design a control function on the preset page to stop the display of unwanted information that is on the component list of the VSCI.

**Visiting difference of clusters based on the preliminary survey.** We also checked whether there were visiting differences by cluster based on the preliminary survey regarding the degree of interest in VSCI elements. Interestingly, we did not find any significant differences in either behavioral factors (visiting time: $t(78) = 0.42$, $p = 0.43$; viewing time for artwork: $t(78) = 0.32$, $p = 0.52$; viewing time for app: $t(78) = 0.35$, $p = 0.60$) and satisfaction (exhibition satisfaction: $t(78) = -0.47$, $p = 0.64$; app satisfaction: $t(78) = 0.124$, $p = 0.902$). While the online survey showed notable differences in participant curiosity for all items, no differences in information preference were found for almost all VSCI factors. Unlike the participants in Cluster 2 who negatively evaluated the VSCI factors presented through the questionnaire, they had a more positive evaluation than they expected when they viewed the actual VSCI data in the exhibition environment (e.g., P17, "It was better to see information from other visitors than I previously thought. I was able to think more deeply by looking at other people's thoughts and scores on the artwork"). However, the only difference was in "Comments on artworks" ($Mean_{diff}(cluster1, cluster2) = 0.6$; $t(38) = 2.30$; $p = 0.03$). The participants in Cluster 2 answered negatively regarding the visitors' comments because this information was of low quality, subjective, and unprofessional (e.g., P13, "I have doubts about the professionalism of those who provided the comments. I think it would be good if the opinion came from an expert, to have credibility.") Although we did not get any significant or notable findings, the result revealed that the level of professionalism conveyed by the information also needs to be considered in audience-based information.

## Discussion

The supplemental information provided by museums has been focused on delivering expert knowledge about artists or artworks to visitors. In addition, the analysis of various visitor reactions has been used for exhibition evaluation or research purposes. Going beyond these previous approaches, this study proposed a way to provide visitors with others' responses to an art exhibition via a mobile guidance application. Specifically, through a visitor experiment, we revealed that providing VSCI has the potential to improve visitor exhibition experience in terms of visitor behavior and satisfaction. From the interviews, we found that visitors who compared their thoughts with others gained new insights on art appreciation through different opinions and found key points in viewing the exhibition via VSCI.

We also identified visitors' additional needs for VSCI. First, many participants look forward to opportunities for interaction that are designed based on VSCI, such as sharing their visitor type results on SNS, replying to or liking others' feedback, having ranking games based on visiting behavior, pairing people with similar visiting types to meet up for a museum trip, or other personalized functions. Second, personalization was often mentioned, for example, having a personal archive page for visitors' feedback records or providing visit recommendations based on individuals' visit-related reactions. Third, participants also suggested that the options for emotional responses could be more varied (e.g., P11, "In particular, the items of emotion for artworks were strange. It would be nice if less extreme emotions, such as peaceful and joyful, were taken into account"). In this study, we applied Ekman's model, and it is necessary to

consider the list of emotions presented in emotional studies related to viewing art, such as those used by Rodriguez-Boerwinkle et al. [89]. In summary, we noticed an interest in enriched interactive contents with VSCIs, indicating that future research on this topic is warranted.

This study focused mainly on mobile guidance applications, and other applications that use VSCI in various museum service areas should be considered for comparison. For instance, to recall the audience's visit, Petrelli et al. [90] suggested providing tangible data souvenirs by capturing the personal visiting experience. If visitor-contributed information from other visitors, such as VSCI, is combined with the function of generating data souvenirs, more diverse interactive services can be created to extend the visitor experience in art museums. VSCI can also be offered to online visitors in virtual museums. Walmsley [91] demonstrated that the digital engagement of visitors could enhance the opportunities to attend to artistic dialogue more frequently and encourage empathy with others in terms of sociological roles. Sundar et al. [92] revealed that applying communication technology (customization of the gallery, interactivity through live chats, and 3D navigational tool) in virtual museums can positively support the quality of visitors' experience. Moreover, social connectivity between online visitors via VSCI can be linked to the metaverse, which has received growing interest and represents a future trend of virtual museums [93]. Therefore, it is necessary to investigate the use of VSCI in various museum service areas.

Lastly, one of the impressive interview answers was that just giving various information elements made a visitor think more about the artwork (P21, "I liked the fact that I got to think about my own thoughts, rather than the results of others' responses while looking at the information elements, for example, what emotions I felt, what features of artwork were good, and why this artwork was good for me"). Regarding individual interpretations of artworks that are important in art education [94], it is meaningful to identify information elements that help visitors independently think and appreciate artworks (as in the VSCI), rather than simply using elements that have been commonly discussed in visitor studies.

## Limitations and future research

This study has several limitations that need to be interpreted carefully. First, as in previous empirical studies [95], the experiment was conducted in a laboratory environment with a small number of participants. However, because of the spread of COVID-19, it was difficult to conduct experiments in a museum environment and recruit a large number of participants. Second, even though the purpose of the online survey in Stage 1 was finding insights to design a mobile guidance application with VSCI, the sample size was quite small (n = 71). In addition to the purpose of understanding user demand, in-depth research is required in consideration of various visitor characteristics (e.g., age differences [39], the composition of the visit groups [96], and countries [97]) to derive standardized VSCI.

Third, a small number of artworks were installed in the experimental exhibition (ten artworks for each exhibition). Meanwhile, large art museums usually take a relatively long time to visit because of the huge number of displayed artworks. In such an environment, visitors can experience museum fatigue [98, 99], and providing more information, such as VSCI, may accelerate visitors' fatigue. However, because VSCI captures artworks that visitors pay attention to, such social recommendations could reduce the museum fatigue phenomenon. Hence, it is necessary to examine the effect of VSCI in a real museum environment on a large-scale basis in future studies. Lastly, through the online survey, we found that the respondents can be divided into two groups based on how they viewed information on other visitors' reactions: positive or negative. Therefore, we examined the characteristics of the two groups through an

experiment, but no significant difference was found in all aspects of behavioral factors and satisfaction. Based on previous studies on visitor classification, this topic may be further investigated by grouping visitors based on behavioral characteristics in the exhibition [100, 101], visit motivation [102], and anticipated experience [2].

## Conclusions

This study investigated the effect of providing information from visitors via mobile guidance applications on museum visitors' experiences. The contributions of this research are twofold. For academic interest, there is the need for an in-depth understanding of visitor-oriented social information in the field of visitor studies. To do this, based on previous visitor studies, we defined VSCI as encompassing the behaviors, emotions, and opinions of visitors. We then developed a mobile application for VSCI and conducted a visitor experiment using mobile eye-tracking technology. The experiment results demonstrated that when VSCI was provided, the behavioral parameters of the visitors increased, and visitor satisfaction for the exhibition and the app also improved. Moreover, the interview analysis indicated that the exhibition experiences of visitors were enriched through VSCI. This means that the information elements that previous studies have focused on can be used as data sources for researchers and museum operators and as information sources for museum visitors. To our knowledge, this is the first attempt to examine visitor-based social information to enhance visitor experience in art museums.

For practical interest, we proposed a method of designing and developing a mobile museum guidance application, including VSCI. This method could help museums use social information as mobile app content. Despite the growing interest in mobile guidance applications, some contemporary museum apps have failed to satisfy visitors' expectations [103]. To address this issue, this study proposed a method for improving visitor satisfaction by paying attention to visitor-contributed information, which has been insufficiently considered in the past. In summary, this study suggested one possible approach of providing social information for the visitor-centered design of museum guidance applications, which contributes to the field of visitor studies. We hope that our research will help enhance visitors' museum experiences by providing them an opportunity to communicate their various opinions and reactions, which goes beyond simply conveying information based on expert knowledge.

## Supporting information

**S1 Appendix. Explanation of online survey questionnaires.**
(PDF)

**S2 Appendix. Explanation of post-experiment interview.**
(PDF)

**S3 Appendix. Explanation of materials for visitor experiment.**
(PDF)

**S4 Appendix. Research data.**
(XLSX)

## Acknowledgments

We thank the Lee-Ungno Art Museum for providing valuable information and Jihye Oh for assisting with the mobile app development.

## Author Contributions

**Conceptualization:** Taeha Yi, Hao-yun Lee, Ji-Hyun Lee.

**Data curation:** Taeha Yi.

**Investigation:** Taeha Yi, Joosun Yum.

**Methodology:** Taeha Yi, Joosun Yum.

**Supervision:** Ji-Hyun Lee.

**Validation:** Taeha Yi.

**Visualization:** Taeha Yi, Hao-yun Lee.

**Writing – original draft:** Taeha Yi, Hao-yun Lee.

**Writing – review & editing:** Taeha Yi, Hao-yun Lee, Ji-Hyun Lee.

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
