## [Decision Letter · Decision Letter 0]

9 Feb 2022

PONE-D-21-36495The influence of visitor-based social contextual information on visitors’ museum experiencePLOS ONE

Dear Dr. Lee,

Thank you for submitting your manuscript to PLOS ONE. After careful consideration, we feel that it has merit but does not fully meet PLOS ONE’s publication criteria as it currently stands. Therefore, we invite you to submit a revised version of the manuscript that addresses the points raised during the review process.

The reviewer has raised several issues concerning the literature review, the methodology, and the results description. Please be sure to address those concerns in your revised version.==============================

We look forward to receiving your revised manuscript.

Kind regards,

Maurizio Naldi

Academic Editor

PLOS ONE

Journal Requirements:

a) Did participants provide their written or verbal informed consent to participate in this study?

Reviewers' comments:

Reviewer's Responses to Questions

**Comments to the Author**

1. Is the manuscript technically sound, and do the data support the conclusions?

Reviewer #1: Yes

2. Has the statistical analysis been performed appropriately and rigorously? 

Reviewer #1: Yes

3. Have the authors made all data underlying the findings in their manuscript fully available?

Reviewer #1: Yes

4. Is the manuscript presented in an intelligible fashion and written in standard English?

Reviewer #1: Yes

5. Review Comments to the Author

Reviewer #1: While the study seems interesting, there are several issues and concerns that should be addressed. I hope the authors find the following comments and suggestions useful.

Introduction

1. Why is there a need to focus more on the interactions between personal and social contexts? The research justification was unclear.

2. Social information has been widely studied. Why is there a need for this study? Please justify the research question/gap/significance/contribution.

3. The research contributions were unclear. What does the study provide with respect to others?

Literature Review

4. On page 4 lines 67-68, “However, research using various data collected to benefit visitors (rather than museum experts) is insufficient.”, what did you mean about this?

5. The examples of social information on pages 4-5 were unclear.

6. Please define each variable/concept of the study (conceptualization and operationalization) in the literature review.

7. Research questions/hypotheses were not well presented in the literature review.

8. It was unclear how you determined and proposed the elements of VSCI. A conceptual framework discussion is recommended.

Methodology

9. I suggest the author(s) have a section to discuss the research design. Is there any reference to support your research design and process?

10. In stage 1, the sample size was small. How did you determine the proper sample size? What were the qualifications of respondents? Were they domestic visitors? How and when did you collect data? Why did you use an online survey since it has several limitations?

11. In stage 3, how did you determine the proper sample size? What were the qualifications of participants? How did you recruit them? When did you conduct the experiment?

12. Where was the discussion of stage 4 in the methodology? Were interviews in stage 4? Who were the interviewees?

Results

13. Often, visitor satisfaction and behavior are affected by museum interpreters or tour guides. Have you considered the impact of the interactions between them? Were they individual/independent visitors or visitors from a package tour?

Discussion and Implications

14. Some findings were consistent with previous studies. What is the difference between the current paper and prior works? Please strengthen the uniqueness of the study.

15. This study failed to answer the “so what” question. Please strengthen the theoretical and practical implications of this study.

6. PLOS authors have the option to publish the peer review history of their article (what does this mean?). If published, this will include your full peer review and any attached files.

Reviewer #1: No

---

## [Author Response · Author response to Decision Letter 0]

16 Mar 2022

Dear Editor and Reviewers, 

We were pleased to have an opportunity to revise our manuscript titled “The influence of visitor-based social contextual information on visitors’ museum experience.” We thank you for reading our manuscript thoroughly and offering constructive feedback. We are very grateful for your comments and have tried our best to address them by commenting below and highlighting the changes made in the revised manuscript. Both our comments below and the changes in the main document are highlighted in purple. We hope we have addressed all the comments and that the manuscript is now suitable for publication in PLoS ONE.

- Editor:

We thank the editor for supporting our work and for the invitation to submit a revised version. 

(1) We added an ethics statement as follows: The participants were provided with a written consent form, and we collected the completed forms. Therefore, we updated “written” in the Procedure section (Line 355)

(2) We updated the reference format for Plos One style.

- Reviewer #1:

First, we are grateful for your detailed review. We tried to enhance the quality of our research paper based on your valuable comments. 

[1] Abstract

1. Why is there a need to focus more on the interactions between personal and social contexts? The research justification was unclear. 

Answer:

We attempted to clarify these statements as follows: 

(a) We revised this sentence, “yet only a few have focused on the interactions” to

“yet the interactions between personal and social contexts have not been fully researched.” 

(b) We also added the following to lines 15–17 in the Abstract:

“Since previous studies related to these interactions have focused on the face-to-face conversation of visitor groups, attempts to provide the social information contributed by visitors have not progressed.”

[2] Introduction

2. Social information has been widely studied. Why is there a need for this study? Please justify the research question/gap/significance/contribution.

Answer:

Currently, most art museums have not been tried to provide social information to visitors of both offline and online exhibitions. In addition, in terms of social contexts, previous studies have focused on face-to-face conversations between visitors and have been limited to group interactions. Therefore, this research investigates the possibilities of providing visitor-contributed social information to visitors and examines the influence of the information on visiting experience. We rewrote some parts of the Introduction and added two references in 46–60 Lines.

(Added references)

[12] Basu C, Hirsh H, Cohen W. Recommendation as classification: using social and content-based information in recommendation. In: Proc 15th Nat Conf on Artificial Intelligence. Madison (1998. pp. 714–720).

[13] Godes D, Silva JC. Sequential and temporal dynamics of online opinion. Marketing Science. 2012. doi:10.1287/mksc.1110.0653

3. The research contributions were unclear. What does the study provide with respect to others?

Answer:

A main contribution of our research is that it reveals the possibility of enhancing the visitor experience via social information, VSCI. Therefore, we tried to update the last part of the Introduction in 61–69 Lines.

[3] Literature Review

4. On page 4 lines 67-68, “However, research using various data collected to benefit visitors (rather than museum experts) is insufficient.”, what did you mean about this?

Answer:

Based on the advancement of technologies, contemporary art museums and researchers have collected visitors’ behavioral and emotional data to understand visitors. However, these data have not been provided to visitors. Our starting point of this research was the question, “Why don’t museums provide visitors with the numerous data collected from visitors?” Subsequently, we revised this in 78–81 Lines. 

5. The examples of social information on pages 4-5 were unclear.

6. Please define each variable/concept of the study (conceptualization and operationalization) in the literature review.

7. Research questions/hypotheses were not well presented in the literature review.

8. It was unclear how you determined and proposed the elements of VSCI. A conceptual framework discussion is recommended.

Answer:

Based on these comments, we realized that our explanation of VSCI has some problems. As you suggested, we added a conceptual framework of VSCI with further explanations. Our revisions are as follows:

(a) We updated the first explanation of VSCI in the literature review and provided two research questions (Lines 98–106).

(b) We changed the section title, “Components of visitor-based social contextual information” to “Conceptualization of visitor-based social contextual information.” (Line 108)

(c) A conceptual framework was added with detailed information (Lines 109–122), and A new figure (Fig.1.) was added.

[4] Methodology

9. I suggest the author(s) have a section to discuss the research design. Is there any reference to support your research design and process?

Answer:

We designed the research process with reference to the prototype-based studies: (1) insight research (or user requirements); (2) app or system design and prototyping; and (3) testing or experiment. As you suggested, we updated the explanation with two additional references [65,66] in Lines 171–182.

(Added references)

[65] Kang B, Song B, Yang S, Lee J. A Development Technique for Mobile Applications Program. Applied Computing and Information Technology 2017 (pp. 47–62). Springer, Cham. doi:10.1007/978-3-319-51472-7_4

[66] Liedtke C, Baedeker C, Hasselkuß M, Rohn H, Grinewitschus V. User-integrated innovation in Sustainable LivingLabs: An experimental infrastructure for researching and developing sustainable product service systems. J Clean Prod. 2015. doi:10.1016/j.jclepro.2014.04.070

10. In stage 1, the sample size was small. How did you determine the proper sample size? What were the qualifications of respondents? Were they domestic visitors? How and when did you collect data? Why did you use an online survey since it has several limitations?

Answer: 

(a) Based on this valuable comment, we found that we should explain this limitation carefully to the reviewer(s) and reader(s). In this study, an online survey was performed to understand the design insights and user demands about social information for mobile museum guidance apps (e.g., Gugenheimer et al.* performed an online survey (n = 48) for gaining user demand). This process corresponds to the initial stage of the method for developing a mobile museum app considering VSCI, and our method can be used by future researchers interested in VSCI. However, the survey results in Stage 1 are not constant, and the probability of the differences could be because of visitors’ characteristics, such as ages, visit groups, or countries. Therefore, we added this limitation in the “Limitations and future research” section with two additional references [96,97] (Lines 650–654). 

* Gugenheimer J, Stemasov E, Frommel J, Rukzio E. Sharevr: Enabling co-located experiences for virtual reality between hmd and non-hmd users. Proceedings of the 2017 CHI Conference on Human Factors in Computing Systems (CHI ‘17). ACM, New York, NY, USA, 4021-4033. doi: 10.1145/3025453.3025683

(Added references)

[96] Sandifer C. Time‐based behaviors at an interactive science museum: Exploring the differences between weekday/weekend and family/nonfamily visitors. Science Education. 1997. doi:10.1002/(SICI)1098-237X(199711)81:6<689::AID-SCE6>3.0.CO;2-E

[97] Rawlings D, Vidal N, Furnham A. Personality and aesthetic preference in Spain and England: Two studies relating sensation seeking and openness to experience to liking for paintings and music. European journal of personality. 2000. doi:10.1002/1099-0984(200011/12)14:6<553::AID-PER384>3.0.CO;2-H

(b) We updated the explanation of Stage 1 in the Methods section (Lines 203–205).

We also added the following clarification in Lines 205–206:

“The survey responses were collected via Google Forms from May 24 to June 8, 2021.”

And in Line 216:

“from multiple online communities in South Korea (e.g., ARA and Hongik-in).”

11. In stage 3, how did you determine the proper sample size? What were the qualifications of participants? How did you recruit them? When did you conduct the experiment?

Answer:

(a) The quantitative statistics studies need to “test at least 20 users to get statistically significant numbers (Nielson Norman group, 2012*).” Also, referring to the sample size of these previous studies (Brieber et al., 2014 [95]; Paasschen et al., 2015 [49]) that conducted eye tracking experiments on museum visitors, we determined the number of samples (n = 20) for each condition.

* Nielsen Norman Group (2012). How Many Test Users in a Usability Study? retrieved from https://www.nngroup.com/articles/how-many-test-users/

(b) We revised and updated some parts of Stage 3, as follows (Lines 329–330):

“We contacted those who agreed to participate in this experiment via the online survey.”

(Error found) “male:female = 19:21” in Line 331.

“This experiment was conducted from August 6 to August 21, 2021, as three blocks (Fig 4) and took about 90 mins, on average.” (Lines 353–354)

12. Where was the discussion of stage 4 in the methodology? Were interviews in stage 4? Who were the interviewees?

Answer:

We found the statement problems in describing the post-experiment interview. The interviews were conducted in the third block of Stage 3 (Fig. 4; see 368–378 lines); however, our figures did not describe it. Therefore, we changed the two Figures (Fig. 2 and Fig. 4) and some parts of the manuscript as below:

(a) “four stages” to “three stages” in Line 179.

(b) Changing Figure 2 and 4 to mention the post-experiment interview in Line 198 (Fig. 2) and Line 377 (Fig. 4);

(c) Adding “with a post-experiment interview” in Line 190.

[Results]

13. Often, visitor satisfaction and behavior are affected by museum interpreters or tour guides. Have you considered the impact of the interactions between them? Were they individual/independent visitors or visitors from a package tour?

Answer:

(a) When VSCI was delivered to individual visitors through the mobile museum guidance application, we paid attention to the impact on their museum experience. (e.g., Line 621: “This study focused mainly on mobile guidance applications”). Therefore, the docent program was not considered in this study.

(b) In the visitor experiment, all participants were independent visitors. As you commented, the visiting groups can have different experiences compared to the VSCI. For this reason, we updated this information in the Limitations and Future research sections (Lines 652–654).

[Discussion and Implications]

14. Some findings were consistent with previous studies. What is the difference between the current paper and prior works? Please strengthen the uniqueness of the study.

Answer:

We understand this concern. We found two parts that do not mention the differences compared to previous works. We revised these two parts (Lines 486–491 and Lines 561–565).

15. This study failed to answer the “so what” question. Please strengthen the theoretical and practical implications of this study.

Answer:

As you suggested, we strengthened the theoretical and practical implications or contributions in three points:

(a) For academic interest (Lines 672–674):

“The contributions of this research are twofold. For academic interest, there is the need for an in-depth understanding of visitor-oriented social information in the field of visitor studies.”

(b) For practical interest (Lines 685–687):

“For practical interest, we proposed a method of designing and developing a mobile museum guidance application, including VSCI. This method could help museums use social information as mobile app content.”

(c) Additional summary (Lines 691–693):

“In summary, this study suggested one possible approach of providing social information for the visitor-centered design of museum guidance applications, which contributes to the field of visitor studies.”

---

## [Decision Letter · Decision Letter 1]

29 Mar 2022

The influence of visitor-based social contextual information on visitors’ museum experience

PONE-D-21-36495R1

Dear Dr. Lee,

We’re pleased to inform you that your manuscript has been judged scientifically suitable for publication and will be formally accepted for publication once it meets all outstanding technical requirements.

Kind regards,

Maurizio Naldi

Academic Editor

PLOS ONE

Additional Editor Comments (optional):

Reviewers' comments:

Reviewer's Responses to Questions

**Comments to the Author**

1. If the authors have adequately addressed your comments raised in a previous round of review and you feel that this manuscript is now acceptable for publication, you may indicate that here to bypass the “Comments to the Author” section, enter your conflict of interest statement in the “Confidential to Editor” section, and submit your "Accept" recommendation.

Reviewer #1: All comments have been addressed

2. Is the manuscript technically sound, and do the data support the conclusions?

Reviewer #1: (No Response)

3. Has the statistical analysis been performed appropriately and rigorously? 

Reviewer #1: (No Response)

4. Have the authors made all data underlying the findings in their manuscript fully available?

Reviewer #1: (No Response)

5. Is the manuscript presented in an intelligible fashion and written in standard English?

Reviewer #1: (No Response)

6. Review Comments to the Author

Reviewer #1: (No Response)

7. PLOS authors have the option to publish the peer review history of their article (what does this mean?). If published, this will include your full peer review and any attached files.

Reviewer #1: No

---

## [Editor Report · Acceptance letter]

1 Apr 2022

PONE-D-21-36495R1 

The influence of visitor-based social contextual information on visitors’ museum experience 

Dear Dr. Lee:

I'm pleased to inform you that your manuscript has been deemed suitable for publication in PLOS ONE. Congratulations! Your manuscript is now with our production department. 

Kind regards, 

on behalf of

Professor Maurizio Naldi 

Academic Editor

PLOS ONE